# Influence of Finite Mobilities of Triple Junctions on the Grain Morphology and Kinetics of Grain Growth

**Ernst Gamsjäger [1],\*, Boris Gschöpf [1] and Jiří Svoboda [2]**

[1] Institute of Mechanics, Montanuniversität Leoben, Franz-Josef-Straße 18, A-8700 Leoben, Austria; boris.gschoepf@stud.unileoben.ac.at

[2] Institute of Physics of Materials, Academy of Sciences of the Czech Republic, Žižkova 22, 616 62 Brno, Czech Republic; svobj@ipm.cz

\* Correspondence: e.gamsjaeger@unileoben.ac.at; Tel.: +43-3842-402-4006

**Abstract:** Grain boundary networks composed of equal microstructural elements were investigated in a recent paper. In this work a more complicated artificial grain topology consisting of one four-sided, two six-sided and one eight-sided grain is designed to further investigate the influence of grain boundary and triple junction mobilities on the kinetics of the system in more detail. Depending on the value of the equal mobility of all triple junctions, the initially square-shaped four-sided grain changes its shape to become more or less rectangular. This indicates that the grain morphology is influenced by the value of the mobility of the triple junctions. It is also demonstrated that a grain arrangement with low mobility triple junctions controlling the kinetics of grain growth enhances growth of the large eight-sided grains. In addition, grain growth is investigated for different values of mobilities of triple junctions and grain boundaries. A strong elongation of several grains is predicted by the modeling results for reduced mobilities of the microstructural grain boundary elements. The two-dimensional modeling results are compared to micrographs of a heat-treated titanium niobium microalloyed steel. This feature, namely the evolution of elongated grains, is observed in the micrograph due to the pinning effect of (Ti, Nb)C precipitates at elevated soaking temperatures of around 1100 °C. Furthermore, the experiments show that a broader distribution of the grain sizes occur at 1100 °C compared to soaking temperatures, where pinning due to precipitates plays a less prominent role. A widening of the distribution of the grain sizes for small triple junction mobilities is also predicted by the unit cell model.

**Keywords:** triple junction mobilities; grain topology; vertex model; microalloyed steels

## 1. Introduction

Grain growth is a phenomenon occurring in polycrystalline materials at elevated temperatures that aims to decrease the grain boundary area and thus contributes to minimize the total (Gibbs) energy in the system. The microstructural changes linked to grain growth influence the properties of the materials. Generally, a microstructure consisting of small grains with a uniform size distribution results in mechanical properties which outperform materials with a coarse-grained microstructure. Numerous theoretical and experimental investigations were thus conducted for many decades to better understand the physical processes governing grain growth. As an example the effect of solute drag on migrating grain boundaries and interfaces had been investigated theoretically and experimentally in earlier days, see e.g., [1–5], and remains a hot topic until today, e.g., [6–18].

Two different kinetic regimes were identified and classified as normal or as abnormal grain growth, see e.g., [3,19–21]. Several criteria exist to distinguish between abnormal and normal grain growth. A metrical criterion claims that the rate of the ratio of the abnormal grain with grain radius $R_A$ to

mean grain with radius $R_M$ must be positive, i.e., $\mathrm{d}(R_A/R_M)/\mathrm{d}_t > 0$, see Rios [22]. The distribution of the grain sizes in the system remains unimodal during normal grain growth, whereas a bimodal or multimodal distribution evolves during abnormal grain growth. However, the meaning of $R_M$ is ambiguous in the case of a bimodal size distribution of the grains as pointed out by Kang [20]. Thus, Rios and Glicksman [23] proposed a topological criterion for abnormal grain growth $\mathrm{d}N_A/\mathrm{d}_t > 0$, with $N_A$ being the number of faces of the abnormal grain. Various reasons for the occurrence of abnormal grain growth are reported [19]. Large grains could pre-exist or develop in an erratic manner by the impingement of grains which can contribute to abnormal grain growth. However, Thompson et al. [24] showed that: "If uniform grain boundary energy is the only factor affecting boundary motion, an abnormally large grain in a matrix of normal grains does not grow at a higher relative rate than its neighbors." Abnormal grain growth may occur due to second phase particles distributed in the matrix phase, which partially pin the grain boundaries, see e.g., Rios [25]. Tweed et al. [26] investigated grain growth in samples of aluminum containing alumina particles. They state that promoting anomalous (abnormal) grain growth requires control of the oxide dispersion. Oliveira et al. [27] also observed abnormal grain growth in Eurofer-97 steel. They conclude that: "The combination of a high intrinsic grain boundary mobility, size advantage acquired in the early stages of annealing and the presence of local microstructural instabilities (e.g., dissolution and coarsening of $M_{23}C_6$ carbides) explain abnormal grain growth in Eurofer-97 steel."

As stated by Zöllner and Rios [28] a bimodal size distribution indicating abnormal growth of austenite grains may occur at intermediate soaking temperatures ($\vartheta \approx 1100\ °C$) in microalloyed steels. Normal grain growth is attributed to the microstructure of fine austenite grains at a low soaking temperature ($\vartheta \leq 1000\ °C$) and to coarse unpinned austenite grains at a high soaking temperature ($\vartheta \leq 1200\ °C$). The particles (e.g., NbC or (Ti,Nb)C) pin the grains at the low soaking temperature, however at a high temperature the particles dissolve resulting in a subsequent unpinning of grains. Experimental evidence of these features of the microstructural evolution in microalloyed steels can be found in [29–31].

Whereas grain growth usually tends to become faster with increasing temperature (e.g., in microalloyed steels), according to Rheinheimer et al. [18], [32] a transition to slower grain growth is observed in strontium titanate between $1300\ °C$ and $1390\ °C$. They attribute this behavior to two different types of grain boundaries, a high mobility low temperature and a low mobility high temperature grain boundary. It should be noted that Olmsted et al. [33] have demonstrated by a molecular dynamics simulation that some boundaries may move via a non-activated motion mechanism, which significantly increases low temperature mobility. It is clear from the observations mentioned above that the phenomenon of abnormal grain growth is influenced by elements of the microstructure. These elements are triple junctions and grain boundary lines when a simplified two-dimensional setting is considered and quadruple points, triple lines and grain boundary areas in three dimensions. Triple junctions and quadruple points gain influence on the kinetics of grain growth particularly in nanocrystalline materials [34], but this is also the case for vanishing grains in much coarser microstructures. In addition, solute drag is known to reduce the mobilities of grain boundaries. Furthermore, the pinning of certain structural elements by second phase particles or pores can be effective until the driving force for grain boundary migration is below the attraction force of the particles, see e.g., Gottstein and Shvindlerman [35] and Apel et al. [36].

In our model phenomena like solute drag and particle pinning are taken into consideration by reduced effective mobilities of grain boundaries and/or triple junctions. Rios [37] used Hillert's classical equation for grain growth (see [3]) and modified it by considering a term for the pinning force. He concluded that a higher grain boundary mobility can cause abnormal grain growth in case of a sustained mobility advantage and combined with a pinning force it can develop into an abnormal grain before it outgrows its local environment and loses its mobility advantage. On the one hand, it is not in the scope of these semi-empirical growth laws to obtain information about the grain topology. On the other hand, it is possible to investigate the migration of grain boundaries or even grain boundary

networks by means of vertex models, see e.g., [38–42]. The interplay between the kinetics of grain growth and topological features can be investigated with the latter models. It has been observed experimentally and is confirmed in these works that finite mobilities of triple junctions and even quadruple junctions influence the kinetics. The influence of the surrounding grain topology on the kinetics of grain growth is, however, not yet systematically investigated for systems with finite grain boundary mobilities, distinct grain boundary energies and finite mobilities of triple junctions. To this aim a two-dimensional, simplified grain structure consisting of one four-sided, two six-sided and one eight-sided grain is constructed and subjected to grain growth. Different mobilities are attributed to the grain boundaries of different specific energies and to the triple junctions and eventually the evolution of the grain microstructure is investigated. Thereby, the rate-controlling mechanisms during different grain growth scenarios are revealed.

## 2. Problem Description and Methods

### 2.1. Geometrical Arrangement

In a recently published paper [43] the two-dimensional grain boundary network consisted of a certain "unit cell" composed of a triple junction with three adjacent grain boundaries. This unit cell was periodically repeated in order to fill the space. A larger unit cell allowing for a more complicated microstructure is presented in Figure 1a. This unit cell consists of one four-sided (grain I), two six-sided (grains II and III) and one eight-sided grain IV and can be perpetuated without spaces in the plane. The 16 triple points of the arrangement are delineated in the sketch. It follows from the translational symmetry boundary conditions that some of these triple points are equivalent, e.g., the triple point $T_1$ is equivalent to $T_9$ shifted by the vector $\underline{u}$, see Figure 1b. Similarly, triple points $T_{12}$, $T_{13}$ and $T_{14}$ are equivalent to $T_4$, $T_5$ and $T_6$, respectively. The triple points $T_{10}$ and $T_{11}$ are equivalent to $T_3$ and $T_4$ shifted by the vector $\underline{v}$. The triple points $T_{15}$ and $T_{16}$ are equivalent with $T_6$ and $T_7$ shifted by the vector $(\underline{u} - \underline{v})$. The system contains 19 grain boundaries, where the equivalent grain boundaries follow from analogous considerations. The system is thus fully described by 8 independent triple points $T_1$–$T_8$, and 12 independent grain boundaries 1–12. This means that e.g., the structural element consisting of triple points $T_{11}$ and $T_{10}$ and grain boundary 13 is equivalent to the structural element $T_4$ and $T_3$ and grain boundary 3. The area of the unit cell remains constant during time. The simulated grain growth concludes when the four-sided grain 1 vanishes, i.e., topological transformations are not considered.

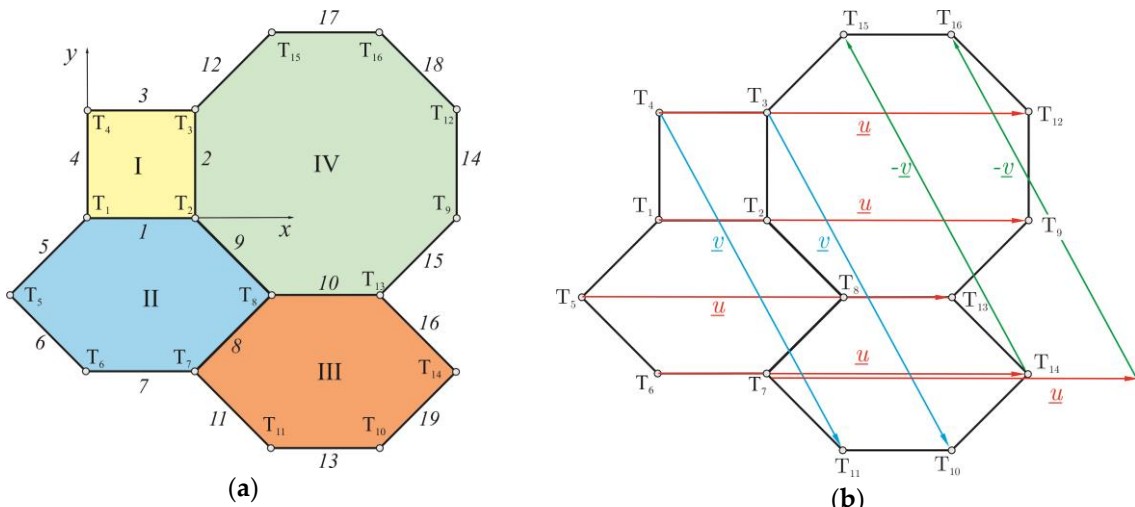

**(a)**　　　　　　　　　　　　　　**(b)**

**Figure 1.** Initial arrangement of the "unit cell" of the grains. (**a**) The four grains are labelled by Roman numerals I-IV. The triple points are denoted as $T_1$- $T_{16}$. The grain boundaries are labelled by Arabic numerals in italics 1-19. (**b**) The displacement vectors u and v are used to construct the triple points $T_9$–$T_{16}$ from triple points $T_1$–$T_7$.

## 2.2. Description of the Numerical Procedure

The kinetics of grain growth is simulated by means of Finite Difference routine programmed in Fortran using an explicit integration scheme. The details concerning the evolution equations and contact conditions are presented in [43]. The grain boundaries are initially discretized by 100 node points. Grain boundaries change their length during grain growth. New node points are generated in case that the distance of the old node points is too far. Node points are eliminated in grain boundary regions with a too high density.

## 3. Results

The kinetics of the arrangement of grain boundaries is investigated, where specific energies $\gamma_{\text{GB }j}$ and individual mobilities $M_{\text{GB }j}$ are assigned to each grain boundary $j$ and individual finite mobilities $M_{\text{T }i}$ of the triple junctions $i$ influence the kinetics of grain growth. The initial length of the side of the quadratic grain is the reference length. The reference time equals the time period for the vanishing of the quadratic grain. The specific energies of the grain boundaries are always set to the same value $\gamma_{\text{GB }j} = 1$, but the product $M_{\text{GB }j}\gamma_{\text{GB }j}$ may be different for each grain boundary due to different individual grain boundary mobilities $M_{\text{GB }j}$. Due to normalizing time and length the mobilities $M_{\text{GB }j}$ and the specific grain boundary energies $\gamma_{\text{GB }j}$ are taken as normalized dimensionless quantities. The normalized triple point mobilities are multiples of the ratio of unit grain boundary mobility to unit length.

### 3.1. From Curvature Controlled to Triple Junction Controlled Grain Growth

In a first example all mobilities $M_{\text{T }i}$ are set to the same value $M_{\text{T}}$, but this value is altered in different simulations. The mobilities $M_{\text{GB }j}$ are all equal and set to a medium value; $M_{\text{GB}} = 1$. It is expected that kinetics of grain growth is controlled by the mobilities of the triple junctions for small values of $M_{\text{T}}$. In Figure 2, the grain arrangement is shown for different normalized times $\bar{t} = t/t_{\text{f}}$, where $t_{\text{f}}$ is the final simulation time when the four-sided grain practically vanishes. The mobilities of the triple junctions are set to a small value, $M_{\text{T}} = 0.1$ in Figure 2a. The evolution of the arrangement for different normalized times $\bar{t}$ is depicted in Figure 2b for $M_{\text{T}} = 200$. This value is sufficiently large, so that the evolution of the arrangement is completely curvature-driven.

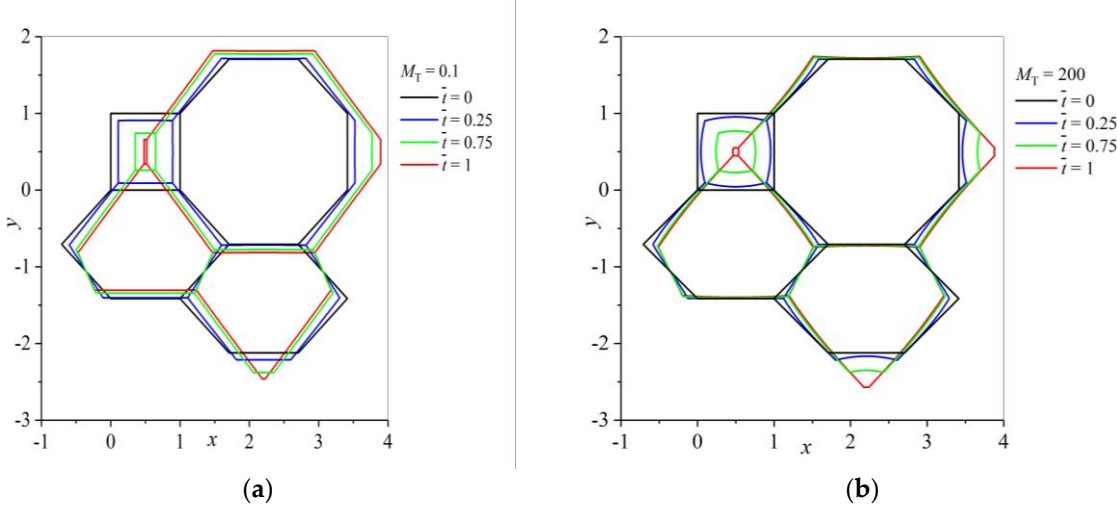

**Figure 2.** Evolution of the grain arrangement for different normalized times $\bar{t}$. (**a**) All mobilities $M_{\text{T }i}$ are set to the small value 0.1. (**b**) All mobilities $M_{\text{T }i}$ are set to a large value 200, so that the kinetics is curvature-driven only.

It is apparent that the grain boundaries in Figure 2a remained almost straight lines and their migration was controlled by small $M_T-$ values. In contrast some of the grain boundaries were distinctly curved in case of high $M_T-$ values. One of the striking differences in the evolution of microstructure is the development of grain I. High triple point mobilities $M_T$ did not significantly change the shape of the initially quadratic grain I. However, small $M_T-$ values resulted in an elongated grain I before the grain vanished. In the following the elongation is described by the *a/b* ratio, where *a* denotes the length of the horizontal sides of grain I and *b* is the vertical length of grain I. The *a/b* ratio is plotted versus normalized time $\bar{t}$ for different $M_T-$ values in Figure 3.

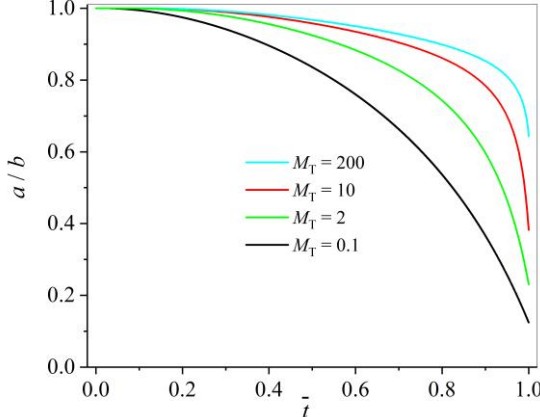

**Figure 3.** The *a/b* ratio of grain I as a function of $\bar{t}$ for different $M_T$ -values.

The triple junction controlled growth followed an almost perfect straight line in the double logarithmic plot, green line presented in Figure 4. The four simulation results with the smallest $M_T-$ values were used for the calculation of the green fitting curve, $t_f = K/M_T$ with the constant K. The fit yielded to $K = (1.375 \pm 0.002)$s. An almost perfect, reciprocal relationship between the time $t_f$ and the mobility $M_T$ was obtained in the range of small $M_T-$ values. In contrast to grain growth controlled by the mobility of the triple junctions, grain growth was dominated by the curvatures of the grain boundaries for sufficiently high triple junction mobilities $M_T$. The horizontal red line marks this limit. Mixed control was observed in the range $0.1 \leq M_T \leq 50$.

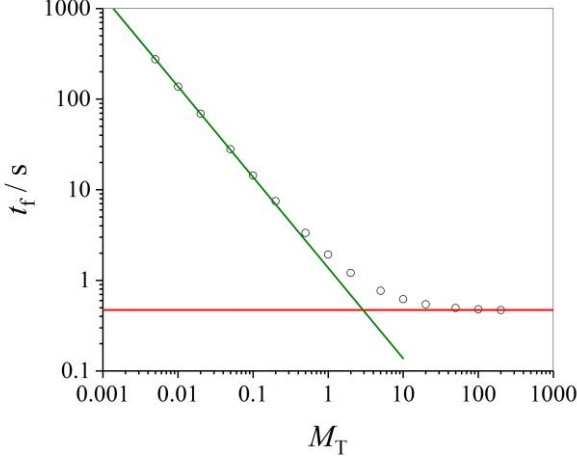

**Figure 4.** The final time $t_f$ is plotted versus the triple junction mobility $M_T$. The dots mark the different simulations for values of $M_T$ ranging from 0.005 to 200. The red line shows the curvature driven regime where the final time remains constant even if $M_T$ is increased. The green line shows the limit of triple point controlled growth.

The evolution of the areas of grains 1–4 are shown for different mobilities $M_T$ in Figure 5. Whereas the areas grew or shrank as linear functions of normalized time $\bar{t}$ in case of curvature driven growth ($M_T \geq 200$), the curves deviated significantly from straight lines for small $M_T-$ values. The areas of the six-sided grains (grain II and III) remained almost constant in case of high $M_T$-values (e.g., $M_T = 200$), but decreased significantly in case of small $M_T-$ values (e.g., $M_T = 0.1$). The 8-sided grain (grain IV) always grew at the expense of the other grains I, II and III. Area of grain IV had the potential to become larger for lower values of $M_T$.

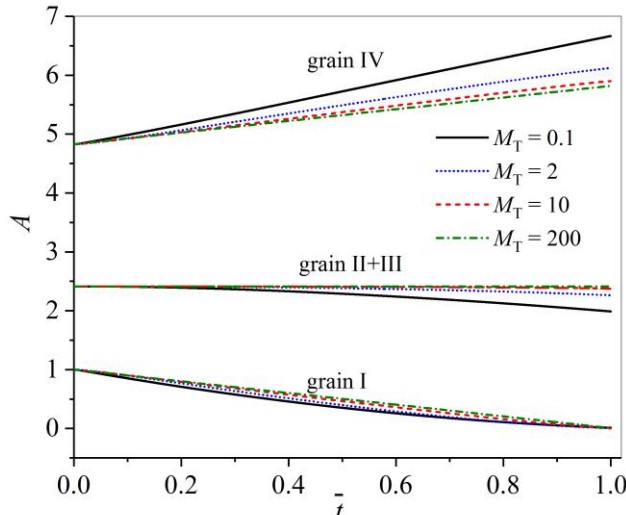

**Figure 5.** Evolution of the areas 1–4 with respect to time at various values of $M_T$.

### 3.2. Structural Elements with Reduced Mobilities

The migration of grain boundaries might be limited by solute drag (e.g., of Nb in steels [44,45]) which can be accounted by a reduced mobility of the grain boundary. Second phase particles might pin grain boundaries, see e.g., Chang et al. [46], where the influence of the morphology of second phase particles on grain growth is investigated utilizing the model by Novikov [47], who considers the drag force of second phase particles. Triple points might also be nearly immobile or experience a reduced mobility. Thus, in the following some case studies are presented where structural elements with reduced mobilities are considered. At first, the mobilities of only certain triple points were reduced and the evolution of the grain network was analyzed. Then, the mobilities of certain grain boundaries were set to smaller values and the influence on the evolution of the structure was investigated. Pinning due to second phase particles was taken into account by setting the mobility of the grain boundary to zero. Finally, network segments consisting of both triple junctions and grain boundaries with reduced mobilities were simulated and analyzed.

### 3.2.1. Triple Points with Reduced Mobilities

In the first case (case I) the mobilities of the triple points 1, 2, 3 and 4 were reduced, $M_{T\,i} = 1$, ($i = 1, 2, 3, 4$), whereas the mobilities of the remaining independent triple points were large $M_{T\,i} = 100$, ($i = 5, 6, 7, 8$), see Figure 6a. In the second case (case II) the mobilities of the triple points 5, 6, 7 and 8 were reduced, $M_{T\,i} = 1$, ($i = 5, 6, 7, 8$), whereas the mobilities of the remaining independent triple points were large $M_{T\,i} = 100$, ($i = 1, 2, 3, 4$) see Figure 6b.

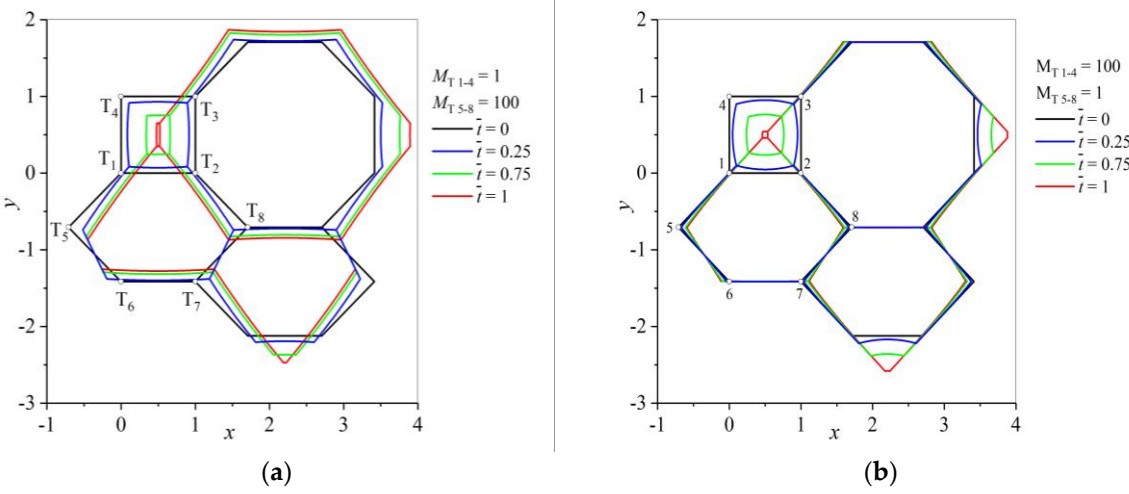

**Figure 6.** Evolution of the grain arrangement for different normalized times $\bar{t}$. (**a**) Case I: The mobilities $M_{\mathrm{T}\,1\text{–}4}$ are set to the comparatively small value 1, $M_{\mathrm{T}\,5\text{–}8}$ are set to 100. (**b**) Case II: The mobilities $M_{\mathrm{T}\,5\text{–}8}$ are set to the comparatively small value 1, whereas, $M_{\mathrm{T}\,1\text{–}4}$ are set to 100.

It is evident that the evolution of the grain arrangement depends on the position of the triple points with reduced mobility. The area of the eight-sided grain became considerably larger in case I (Figure 6a) compared to case II (Figure 6b). The reduced mobility of the triple points of grain I (case I) resulted in an elongation of the initially quadratic grain, whereas the a / b ratio remained almost constant in case II. The grain boundaries that separated grain I from grain II and grain I from grain III, respectively were considerably displaced and shortened during the evolution in both cases I and II. All other grain boundaries of grain II and III were almost immobile due to reduced triple point mobilities $M_{\mathrm{T}\,i} = 1$, $(i = 5, 6, 7, 8)$ in case II; grain II and grain III were strongly deformed in case I. Grain IV was growing at the expense of all remaining grains in case I, while this grain IV was growing mainly at the expense of grain I in case II. Grain I became strongly elongated before it vanished for the conditions used in case I. In contrast, the shape of grain I was almost conserved for case II.

In the next case III the mobility of triple point 2 was reduced, whereas all other triple point mobilities remained high. In case IV the mobility of triple points 2 and 8 were reduced. Comparison of cases III and IV revealed that the evolution of the grain arrangement was strongly influenced only by reducing the mobility of the triple points belonging to the four-sided grain. It is shown in Figure 7a (case III) that a reduced mobility of triple point 2 resulted in a strongly distorted grain boundary structure. A reduced mobility of triple point 8 did not significantly influence the topology of the grain structure, see Figure 7b (case IV). Thus, the grain structures (compare Figure 7a,b) appeared to look similar before grain I disappeared.

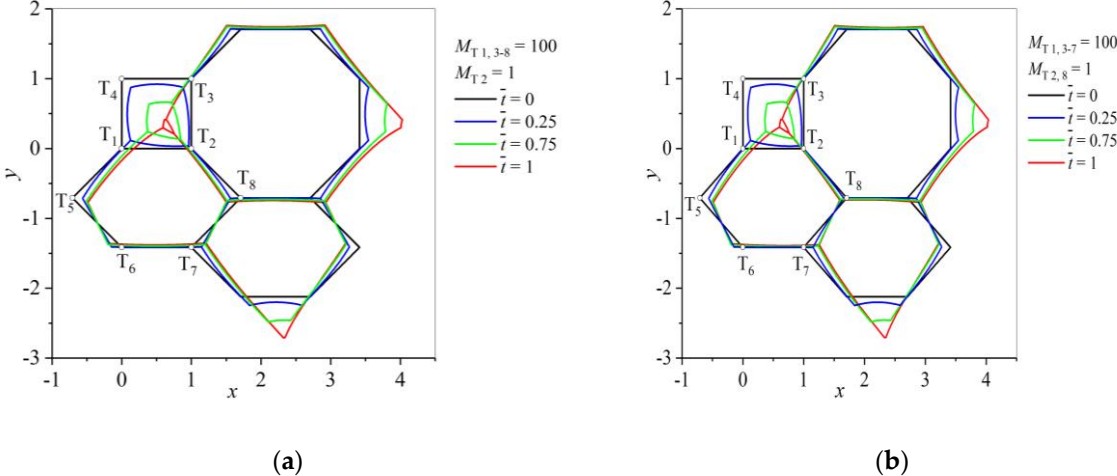

(**a**)　　　　　　　　　　　　　　　　　　　　(**b**)

**Figure 7.** Evolution of the grain arrangement for different normalized times $\bar{t}$. (**a**) Case III: The mobility $M_{T\,2}$ is set to 1, all other triple point mobilities $M_{T\,1,3-8}$ are set to 100. (**b**) Case IV: The mobilities $M_{T\,2,8}$ of the triple points 2 and 8 are set to 1, whereas, the $M_{T\,1,3-7}$ of the remaining triple points are set to 100.

The mobilities of the triple points 1 and 2 were reduced in case V (Figure 8a) and the mobilities of the triple points 2 and 3 were reduced in case VI (Figure 8b). All mobilities of the other triple points were set to high values. The shrinkage of grain I exhibited different features when cases V and VI were compared. Grain I with blue grain boundaries of at normalized time $\bar{t} = 0.25$ looked to be rotated by an angle of 90° in anticlockwise direction if one compares the evolution of grain I shown in Figure 8a,b, respectively. However, at higher normalized times the grain boundary between triple points 1 and 4 shrank more in case V than in case VI. Grain I disappeared with an almost equilateral triangle shape at a normalized time $\bar{t} = 1$ in case V, whereas grain I at $\bar{t} = 1$ exhibited the shape of a vertically elongated rectangle in case VI. This is a further indication that the mobilities of the individual microstructure elements play a decisive role in grain growth.

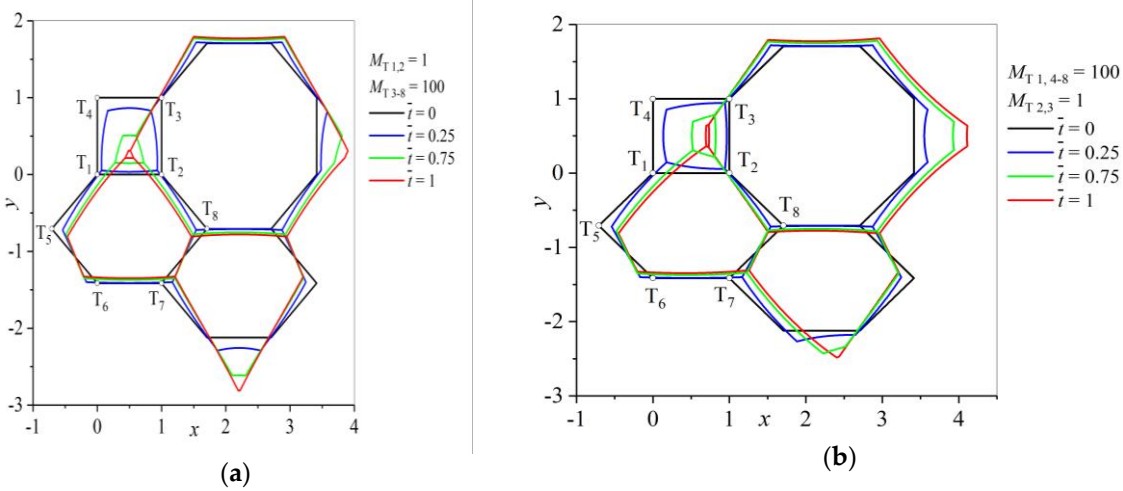

(**a**)　　　　　　　　　　　　　　　　　　　　(**b**)

**Figure 8.** Evolution of the grain arrangement for different normalized times $\bar{t}$. (**a**) Case V: The mobilities $M_{T\,1,2}$ of the triple points 1 and 2 are set to 1, all other triple point mobilities $M_{T\,3-8}$ are set to 100. (**b**) Case VI: The mobilities $M_{T\,1,2}$ of the triple points 2 and 3 are set to 1, all other triple point mobilities $M_{T\,1,4-8}$ are set to 100.

In case VII the triple points of grain I were set to entirely different values (Figure 9a). This resulted in a migration of the grain boundary network that a pretended weak rotation of grains III and IV was observed. Grain IV was able to grow markedly on the expense of the other grains. It is likely that the

irregular structure of grain boundaries observed in grain growth experiments arises from different mobilities of the structural elements. This interpretation is supported by the results of case VII. The triple points 1–4 of grain I are pinned in case VIII (Figure 9b). In this case all grains remained in the structure. The system tended to a constrained equilibrium. The times given in Figure 9b have the following meaning: time $\bar{t}_1$ is the initial time; $\bar{t}_1 = 0$. The final time $\bar{t}_4$ was set to a sufficiently high value so that the grain boundaries did not migrate anymore and $\bar{t}_2$ and $\bar{t}_3$ were chosen in such a way that grain IV increased its area change by 25% and 75%, respectively. The initial angle $\alpha$ was located between a horizontal line from triple point 3 to the grain boundary that contained triple point 3 and separated grain IV and grain III. This initial angle $\alpha = \pi/4$ became $\alpha\prime = \pi/3$ after equilibration.

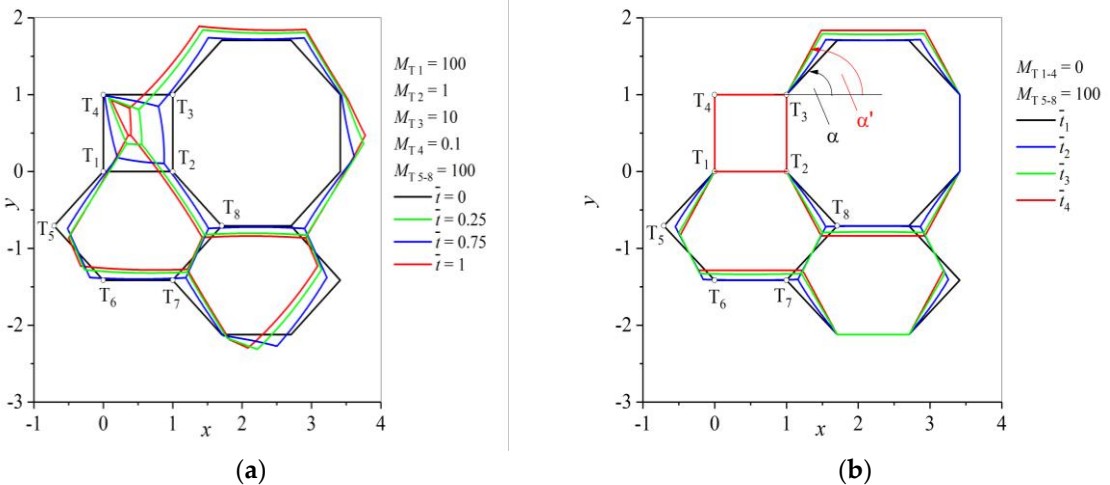

(a)                          (b)

**Figure 9.** Evolution of the grain arrangement for different normalized times $\bar{t}$. (**a**) Case VII: all triple point mobilities not adjacent to grain I are set to high values, and the triple point mobilities of grain I are all distinctly different: Mobility $M_{T\,1} = 100$, $M_{T\,2} = 1$, $M_{T\,3} = 10$ and $M_{T\,4} = 0.1$ set to 1, all other triple point mobilities are set $M_{T\,1,5–8}$ are set to 100. (**b**) Case VIII: The triple points 1-4 of grain I are pinned.

### 3.2.2. Grain Boundaries with Reduced Mobilities

The effect of grain boundaries with reduced mobilities on the evolution of the grain arrangement was investigated in this subsection. The mobilities of all triple points were set to a high value, here 100, so that the influence of the triple junctions on the kinetics of grain growth was negligible. However, certain reduced grain boundary mobilities influenced the evolution of the grain arrangement. In this sense the mobility of certain grain boundaries were set to small values; here the mobility of grain boundary 1 was set to 0.1; $M_{GB\,1} = 0.1$, the remaining grain boundary mobilities are set to 1; $M_{GB\,2–12} = 1$ in case IX, see Figure 10a. In case X the mobility of grain boundary 4 was reduced; $M_{GB\,4} = 0.1$, see Figure 10b, the mobilities of the other grain boundaries are set to 1; $M_{GB\,1–3,5–12} = 1$. In case IX a reduced mobility was attributed to grain boundary 1 and as a consequence grain I became vertically elongated with grain boundary 3 disappearing first. When the mobility $M_{GB\,4}$ was reduced as in case X grain I became horizontally elongated. The vertical elongation of grain I was more pronounced in case X than the horizontal elongation in case IX.

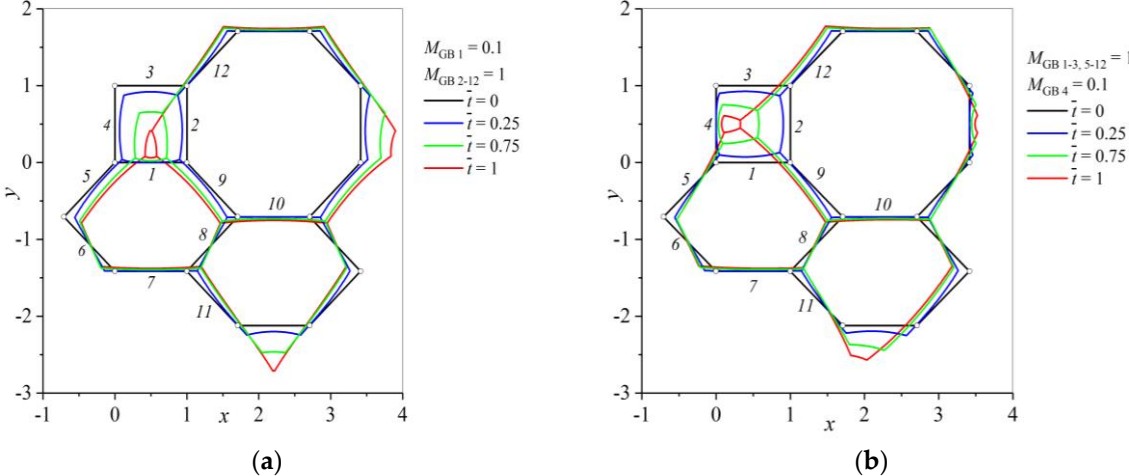

**Figure 10.** Evolution of the grain arrangement for different normalized times $\bar{t}$. (**a**) Case IX: The mobilities $M_{\text{T } 1-8}$ are set to value 100, whereas the mobility of grain boundary $M_{\text{GB } 1}$ is set to a comparatively small value 0.1 and the remaining grain boundary mobilities $M_{\text{GB } 2-12}$ are set to 1. (**b**) Case X: The mobilities $M_{\text{T } 1-8}$ are set to value 100, grain boundary mobility $M_{\text{GB } 4}$ is set to 0.1 and the remaining grain boundary mobilities $M_{\text{GB } 1-3,5-12}$ are set to 1.

In the following two cases XI (Figure 11a) and XII (Figure 11b) the boundary mobilities of the grain boundaries of whole grains were reduced. In case XI the mobility of the grain boundaries of the eight-sided grain were reduced, i.e., the mobilities of grain boundaries 2, 4, 5, 7 and 9–12 were set to 1: $M_{\text{GB } 2,4,5,7,9-12} = 1$, the remaining grain boundary mobilities are set to 10. In case XII the mobility of the grain boundaries of the four-sided grain were reduced, i.e., the mobilities of grain boundaries 1–4 were set to 0.1; $M_{\text{GB } 1-4} = 0.1$, the remaining grain boundary mobilities equal 1.

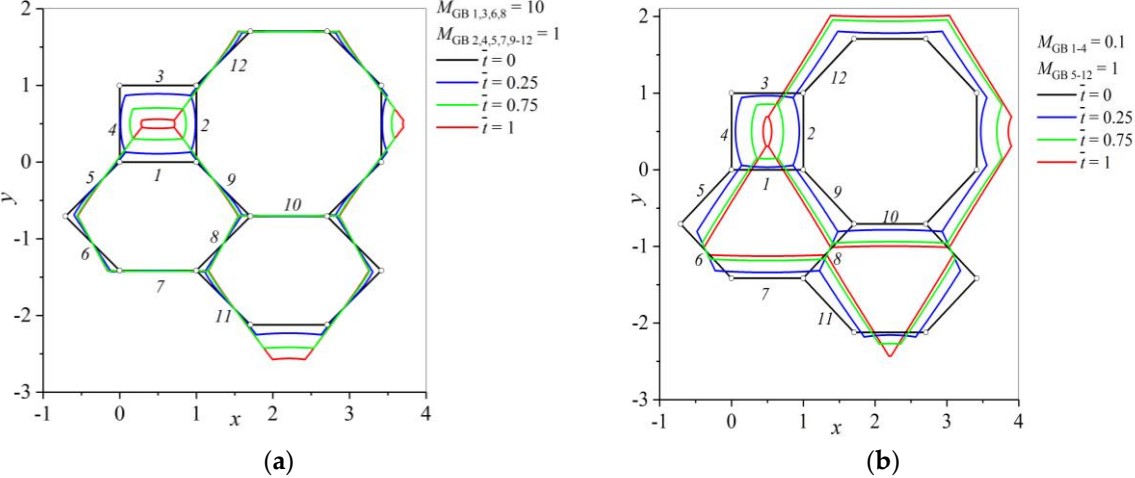

**Figure 11.** Evolution of the grain arrangement for different normalized times $\bar{t}$. (**a**) Case XI: The mobilities $M_{\text{T } 1-8}$ are set to value 100, whereas the mobilities of grain boundaries from the eight-sided grain IV $M_{\text{GB } 2,4,5,7,9-12}$ are set to 1 and the remaining grain boundary mobilities $M_{\text{GB } 1,3,6,8}$ are set to 10. (**b**) Case XII: The mobilities $M_{\text{T } 1-8}$ are set to value 100, grain boundary mobilities of the four sided grain I $M_{\text{GB } 1-4}$ are set to 0.1 and the remaining grain boundary mobilities $M_{\text{GB } 5-12}$ are set to 1.

In cases that reduced mobilities were assigned to the grain boundaries of the eight-sided grain, the two six sided grains II and III grew to a larger extent at the expense of the four-sided grain than grain IV did. The shape of grain I remained symmetric during its evolution with respect to both the horizontal and the vertical direction in both cases XI (Figure 11a) and XII (Figure 11b). The shape of

grain I became horizontally elongated in case XI and vertically elongated in case XII. Grain IV was only slightly growing due to the reduced mobilities of its grain boundaries in case XI, the area of grain IV increased strongly in case XII.

The mobility of grain boundary 8 was reduced; $M_{GB\ 8} = 0.1$ in case XIII (Figure 12a). In case XIV (Figure 12b) the mobilities of grain III were reduced; $M_{GB\ 3,6,8,10-12} = 0.1$. The mobilities of the remaining grain boundaries were higher by one order of magnitude in case XIII and XIV.

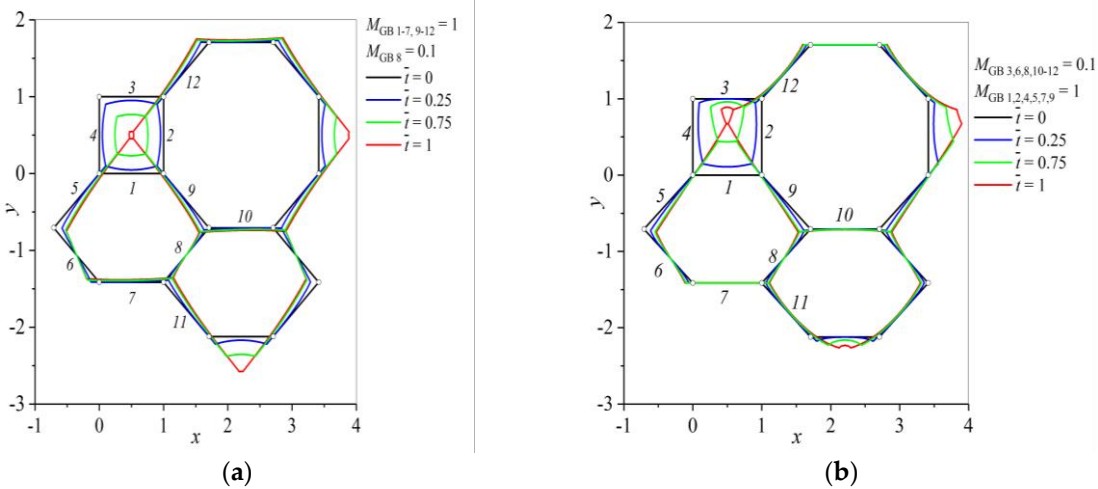

<center>(<b>a</b>)　　　　　　　　　　　　　　　　　　　　　　　　　　(<b>b</b>)</center>

**Figure 12.** Evolution of the grain arrangement for different normalized times $\bar{t}$. (**a**) Case XIII: The mobility of grain boundary 8 is reduced; $M_{GB\ 8}$ is set to 0.1 and the remaining grain boundary mobilities $M_{GB\ 1-7,9-12}$ are set to 1. (**b**) Case XIV: The mobilities $M_{T\ 1-8}$ are set to value 100, grain boundary mobilities of the six sided grain III $M_{GB\ 3,6,8,10-12}$ are set to 0.1 and the remaining grain boundary mobilities $M_{GB\ 1,2,4,5,7,9}$ are set to 1.

It is observed that a reduced boundary mobility $M_{GB\ 8}$ of grain boundary 8 had a marginal influence on the evolution of the grain arrangement. In contrast, the shapes of grain II and III evolved in a completely different manner in case XIV. Grain II strongly grew at the expense of grain I whereas the area of grain III increased only slightly. It is worth mentioning that in the grain boundary network of case XIV some grain boundaries became elongated and strongly curved (see grain boundary 11 and 12 in Figure 12b), whereas others (6, 7, 8 and 10) remain almost straight.

### 3.2.3. Pinning of Microstructural Entities

In the following the evolution of the microstructure was investigated for certain completely pinned grain boundary regions. The mobility of triple point 8, $M_{T\ 8}$, and the mobilities $M_{GB\ 8-10}$ of the grain boundaries 8, 9 and 10 were set to zero in case XV. The mobilities of the remaining triple points $M_{T\ 1-7}$ were set to 100, and the mobilities of the remaining grain boundaries $M_{GB\ 1-7,11-12}$ were set to 1 in case XV (Figure 13a). In case XVI the mobility of triple point 3, $M_{T\ 3}$, and the mobilities $M_{GB\ 2,3,12}$ of the grain boundaries 2, 3 and 12 were set to zero in case XVI (Figure 13b). The mobilities of the remaining triple points $M_{T\ 1,2,4-8}$ were set to 100, and the mobilities of the remaining grain boundaries $M_{GB\ 1,4-11}$ were set to 1. Whereas the microstructure remained symmetric in case XV and the area of grain IV increased only slightly compared to case XVI, the microstructure was heavily distorted in case XVI. The morphology of grain II and grain III evolved differently and lengthened and strongly curved grain boundaries alternated with grain boundaries that remain straight in case XVI.

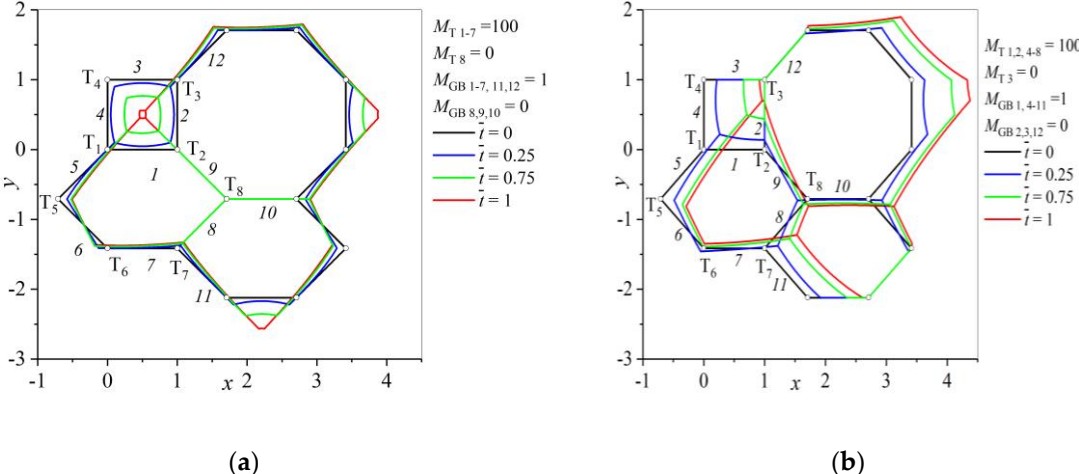

**(a)**                                                                              **(b)**

**Figure 13.** Evolution of the grain arrangement for different normalized times $\bar{t}$. (**a**) Case XV: Triple point mobility $M_{T\,8}$ is set to zero, the mobilities of the remaining triple points are set 100. The mobilities of grain boundaries $M_{GB\,8-10}$ are set to zero and the remaining grain boundary mobilities $M_{GB\,1-7,11,12}$ are set to 1. (**b**) Case XVI: Triple point mobility $M_{T\,3}$ is set to value zero and remaining triple point mobilities are set to 100. Grain boundary mobilities of $M_{GB\,2,3,12}$ are set to zero and the remaining grain boundary mobilities $M_{GB\,1,4-11}$ are set to 1.

It is evident from the computer simulations presented above that the increase of area $A_4$ of grain IV strongly depends on the mobilities of the grain boundaries and triple junctions. The increase of area for different cases is presented in Figure 14. A lower limit was reached when all grain boundaries were immobile except the grain boundaries of grain I. Then grain IV could only gain area at the expense of grain I (dotted line in Figure 14). The increase of $A_4$ was also small in case XV, where the grain boundaries 8, 9 and 10 and their triple point $T_8$ were immobilized (black line). In case that the mobilities of the triple points were set to a sufficiently high value that curvature driven grain growth dominated and the grain boundary mobility was set to 1 the increase of $A_4$ was slightly higher (dash-dotted curve). Compared to this result, the reduction of the mobility of triple point 2 and 3 (case VI) resulted in a higher increase of $A_4$ (red curve). Pinning grain boundaries 2, 3 and 12 (case XV1) leads to a further increase of $A_4$ (blue curve). The highest gain in area in the cases studied occurred for case XII where the mobility of the grain boundaries of the four-sided grain was reduced (green curve). It is worth noting again in this context that the reference time is the time required for the four-sided grain to vanish. Thus, the rate of increase of $A_4$ might have been lower even if the total increase of $A_4$ was higher. This means that the widening of the grain size distribution will occur in systems with reduced grain boundary or triple point mobilities rather than in case of classical curvature mediated grain boundary migration. This must be seen under the precondition that the grain boundary arrangement is kept at the same conditions for the whole process.

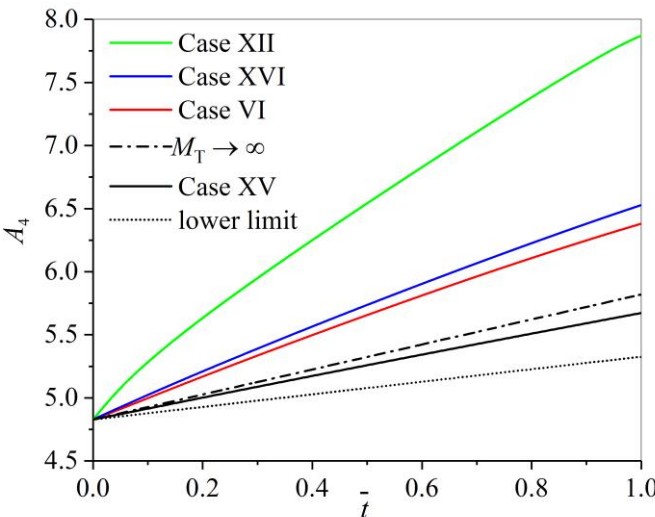

**Figure 14.** Increase of area $A_4$ of grain IV with respect to normalized time for different cases.

Recently, Graux et al. [31] published optical micrographs of steel soaked at 1000 °C, 1100 °C and 1200 °C. While the austenite grains at 1000 °C and 1200 °C seem to be unimodally distributed, the grains at 1100 °C exhibit a deviation from the unimodal distribution. (Ti,Nb)C precipitates are made responsible for pinning austenite grain boundaries. Due to the thermodynamic calculations presented in [31], it can be concluded that the (Ti,Nb)C precipitates are present at 1000 °C and 1100°C and dissolve at 1200 °C and higher temperatures. The microstructures are analyzed by means of software Fiji [48]. As an example the microstructure indicating a bimodal area distribution for thermal etching at 1100 °C is shown in Figure 15a. The grains are approximated by ellipses, where $d$ is defined as the major axis and $c$ is the minor axis (Figure 15b). The $c/d$ ratio is a measure for the deviation of the grains from the circle shape ($c/d = 1$).

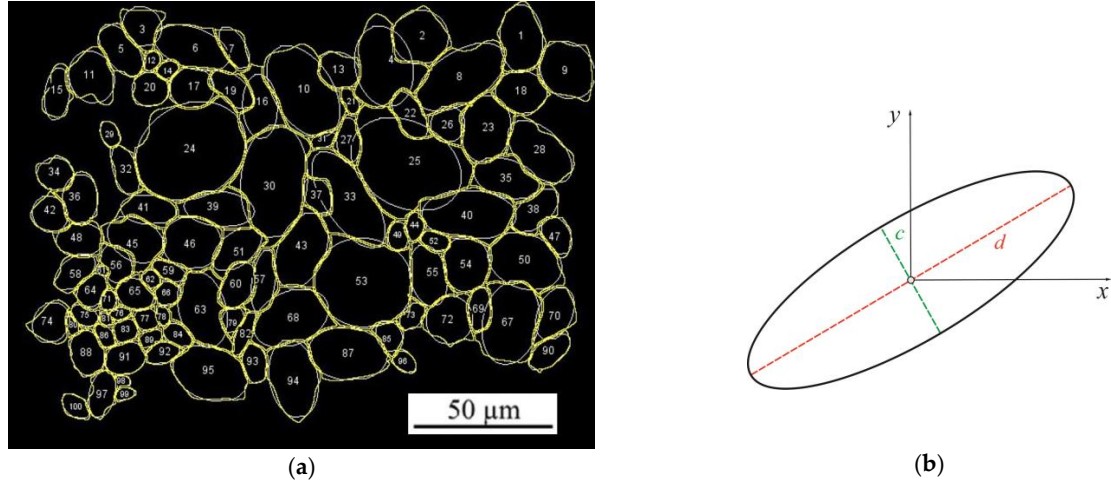

(a)　　　　　　　　　　　　　　　　　　　(b)

**Figure 15.** (**a**) Sectional view of the prior austenite microstructure after thermal etching at 1100 °C (redrawn from [31] and analyzed by Fiji [48]). The grains are approximated by ellipses. (**b**) Ellipse with major axis $d$ and minor axis $c$.

A column plot shows the distribution of the ratio $A_i/A_m$, where $A_i$ is the area of grain $i$ and $A_m$ is the mean area of the grain (Figure 16). The $A_i/A_m$ ratio is represented well for 1000 °C and 1200 °C by fitting with a lognormal distribution. Contrary, this lognormal fit does not work well for 1100 °C, see Figure 16. There are several considerable large grains ($A_i/A_m > 3.5$) at 1100 °C, which are not well described by the lognormal fitting curve.

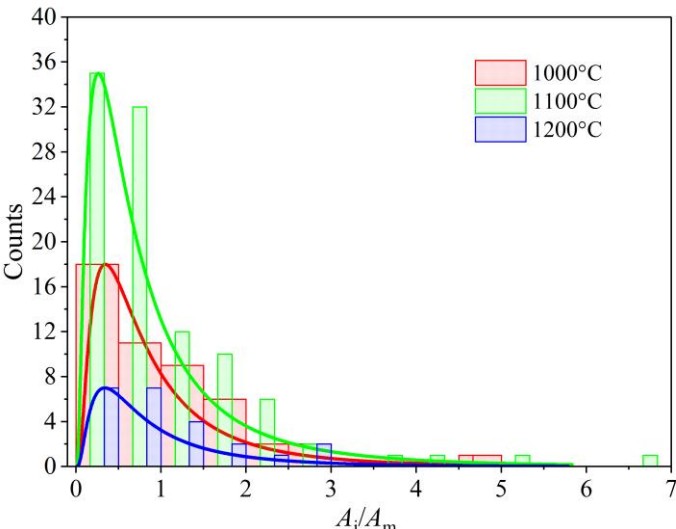

**Figure 16.** Column plots and lognormal distribution curves showing the $A_i/A_m$ ratios for the microstructures etched at 1000 °C (red curve), 1100 °C (green curve) and 1200 °C (blue curve). Original data are taken from [31].

The distributions of these *c/d* ratios are presented in Figure 17. The *c/d* ratios should move from values smaller than 1 to values close to 1 with increasing temperature. This statement holds when the column plots for 1000 °C (red) and 1200 °C (blue) are compared (see Figure 17). However, the green column plots (1100 °C) are shifted to smaller *c/d* ratios.

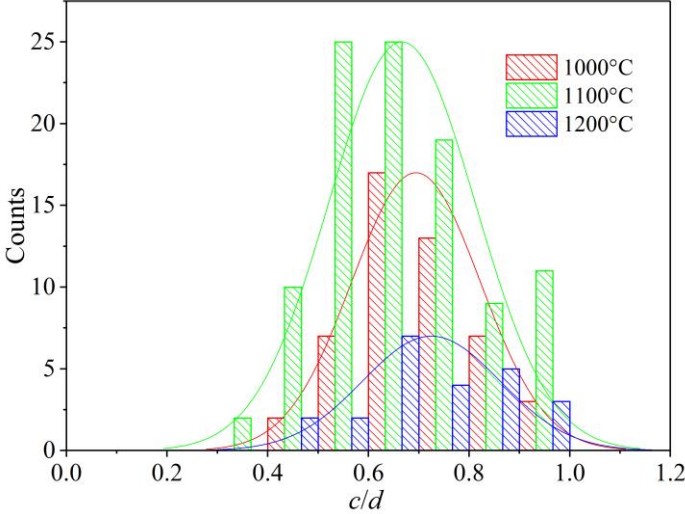

**Figure 17.** Column plots and Gaussian distribution curves showing the *c/d* ratios for the microstructures etched at 1000 °C (red curve), 1100 °C (green curve) and 1200 °C (blue curve).

## 4. Discussion

According to the von Neumann–Mullins law the rate of area change of a certain grain only depends on the number of the neighbors of this grain provided that the grain boundary mobility and the grain boundary energy remain constant. Zöllner and Rios [49] investigated the von Neumann–Mullins relation under triple junction dragging and obtained a similar law for the radius change rate $\dot{R}$ provided that the mobilities of the triple junctions and the grain boundary energies are constant. The change

rate $\dot{a}$ ($a$ is the length of the horizontal grain boundary of four-sided grain I)—similar to $\dot{R}$ in [49]—is proportional to the triple junction mobility in case of triple junction controlled growth of grain I:

$$\dot{a} = -C \cdot M_\text{T}, \tag{1}$$

where $C$ is a positive constant. It follows from integration of Equation (1) that the final time $t_\text{f}$ is inversely related to $M_\text{T}$:

$$t_\text{f} = \frac{a_0}{C} \cdot \frac{1}{M_\text{T}} = \frac{K}{M_\text{T}}, \tag{2}$$

with $a_0$ being the initial length of grain I and with $K$ being a positive constant. It is shown that the two extreme cases—curvature controlled grain growth and grain growth controlled by triple junction dragging—is met in our simulations by plotting the final time $t_\text{f}$ versus different triple junction mobilities $M_\text{T}$ (Figure 4). As the number of neighbors, the grain boundary mobility and the grain boundary energy is kept constant, the time $t_\text{f}$ does not depend on $M_\text{T}$ in case of curvature-driven growth (horizontal red line in Figure 4), i.e., for sufficiently high values of $M_\text{T}$. In contrast the time $t_\text{f}$ is inversely related to $M_\text{T}$ in case of triple junction dragging (green line in Figure 4) and the constant is determined $K = (1.375 \pm 0.002)$s in Section 3.1. Mixed control growth kinetics can be found between these two extreme cases.

The grain boundary energies are always set to a constant value, neither the grain boundary mobilities nor the grain boundary energies are specified in a way that they depend on the misorientation angle between two neighboring grains of the grain boundary. Thus, coincidence site lattice (CSL) grain boundaries are not considered in this work, yet. The different effective grain boundary and triple junction mobilities enable to consider e.g., the effect of alloying elements dragged by grain boundaries and triple junctions in a phenomenological manner. In our future work, where the artificial unit cell might include more grains, it will be a challenging task to take different grain orientations into account, and investigate the influence of certain CSL grain boundaries on the kinetics of grain growth.

In case that the triple junction mobilities $M_\text{T}$ are small, the grain boundaries remain almost straight during their evolution. Since the triple junctions are nearly immobile, the boundaries have enough time to straighten. Although occurring at a slower rate, it has been observed that triple junction mobility controlled grain growth can be more effective with respect to the widening of the grain size distribution (compare Figure 2a,b). The increase of the area of grain IV is investigated for different cases with respect to reduced mobilities of triple junctions and/or grain boundaries (Figure 14). The evolution of the arrangement shows that the area of the two six-sided grains II and III remains almost constant during time for curvature controlled grain growth (Figure 5). However, in case of triple junction dragging, grains II and III shrink resulting in an elongated four-sided grain I and leading to a stronger increase of the area of the eight-sided grain IV. Elongation of grain I, measured by the side ratio $a/b$, increases with decreasing values of $M_\text{T}$ (Figure 3).

It is shown in several examples that the grains may become heavily distorted during their evolution in case that the mobility of certain triple junctions is decreased (Figures 7, 8 and 9a). A similar effect is observed when the mobility of certain grain boundaries is reduced (Figures 10 and 11). Comparable to the results of the computer simulations, heavily distorted grains, straight grain boundaries close to strongly curved grain boundaries and strongly elongated grains can be frequently found in austenite grain structures of micro-alloyed line pipe steels as well (see e.g., [50]). It is expected in these steels that the the effective mobility of grain boundaries and triple junctions is small and inhomogeneously distributed due to solute drag and particle pinning. It is, however, worth noting that even immobilizing certain triple junctions or grain boundaries in the arrangement has almost no effect on the kinetics (Figures 7b and 12a). Thus, it depends markedly on the topological position of the triple junction or grain boundary if its mobility does effect the kinetics or not. It is shown in Figure 13b that pinning of certain structural entities may even give rise to abnormal grain growth. It is worth mentioning that reducing the mobilities of grains with less than 6 sides enhances the growth of the large more than

six-sided grains. It is evident from the green line in Figure 14 that case XII leads to a stronger increase of area of grain IV compared to all other cases investigated.

The area ratio appears to strongly deviate from a unimodally distribution function for a soaking temperature of 1100 °C, whereas 1000 °C and 1200 °C result in unimodally distributed column plots (Figure 16). The calulations (Figure 2) suggest that smaller mobilities of triple points (due to the influence of (Ti,Nb)C precipitates) can be considerd as the onset of abnormal grain growth. The higher temperature and the smaller effective mobilities of the microstructural grain boundary elements is expected to give rise to both, the deviation from the unimodal grain area distribution and the stronger elongation of the grains measured by their *c*/*d* ratios (Figure 17).

## 5. Conclusions

The following conclusions can be drawn based on the results of the computer simulations:

- It is demonstrated that differences in the mobilities of triple junctions and grain boundaries of a grain have the potential to significantly change the shape of this grain. Small differences will always occur e.g., due to spatial fluctuations of dissolved components in the migrating grain boundary or a different retarding forces due to dragging of the triple junctions.
- Even if all triple junctions have the same mobility, the grain arrangement evolves differently depending on the value of this mobility. Grain structures with smaller mobilities of the triple junctions will result in a structure with strongly deformed grains compared to structures, where a smaller influence of triple junction drag on the kinetics prevails. Small mobilities of triple junctions and grain boundaries may enhance the increase of the area of those *n*-sided grains with the highest amount of sides. A broader size distribution can be expected for microstructures with small mobilities of triple junctions rather than in microstructures where curvature-driven grain growth prevails.
- It is demonstrated by simulations that reduced mobilities of microstructural grain boundary elements can give rise to a deviation from a unimodal area distribution and to strongly elongated grains. Supporting these simulation results, titanium and niobium microalloyed steels [31] develop such a microstructure when held at 1100°C due to the retarding force of the (Ti,Nb)C precipitates. The precipitates dissolve at higher temperatures (1200°C) and normal grain growth is obtained eventually.

**Author Contributions:** Conceptualization, E.G.; methodology, E.G.; software, E.G., J.S.; validation E.G., B.G. and J.S.; visualization, E.G.; writing—original draft preparation, E.G., B.G.; writing—review and editing, E.G. and J.S. All authors have read and agreed to the published version of the manuscript.

**Funding:** J.S. gratefully acknowledges financial support provided by the ESIF, EU Operational Programme Research, Development and Education within the research project "Architectured materials designed for additive manufacturing", Reg. No.: CZ.02.1.01/0.0/0.0/16_025/0007304.

**Acknowledgments:** The authors are grateful to Matthias Militzer (University of British Columbia, Vancouver, Canada) for carefully reading and commenting our manuscript. We would also like to express our special thanks to Daiana Cristina Pereira da Silva (Universidade Federal de Ouro Preto, Brazil) for helping us with the FIJI software.

**Conflicts of Interest:** The authors declare no conflict of interest.

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
