# Peer review of "Influence of Finite Mobilities of Triple Junctions on the Grain Morphology and Kinetics of Grain Growth"

_metals, doi:10.3390/met10020185_

Round 1

Reviewer 1 Report

This paper is written at a very high level, I surely suggest to publish in Metals. However, I just have one question to the authors: did you concern your model to CSL boundaries? It seems to me that your model is too general, although there is nothing to critise anything in it. The reference list is correct.

I am not a native English speaker, so it seems to me that the English of the paper is absolutely correct.

I suggest minor revision, which means that I would like to have an answer to my question about CSL boundaries.

Author Response

We want to thank this reviewer for her/his encouraging statements. 

The reviewer asks if we concerned our model to CSL boundaries?

It is possible to consider distinct values for the specific grain boundary energies, when investigating the grain growth with our model. However, in our calculations the grain boundary energies are always set to a constant value, neither the grain boundary mobilities nor the grain boundary energies are specified in a way that they depend on the misorientation angle between two neighboring grains of the grain boundary. Thus, coincidence site lattice (CSL) grain boundaries are not considered in this work, yet. The different effective grain boundary and triple junction mobilities enable to consider e.g. the effect of alloying elements dragged by grain boundaries and triple junctions in a phenomenological manner. In our future work, where the artificial unit cell might include more grains, it will be a challenging task to take different grain orientations into account, and investigate the influence of certain CSL grain boundaries on the kinetics of grain growth.

Reviewer 2 Report

This is a very interesting manuscript dealing with the effect of distinct triple junction mobilities on the morphology of polycrystalline grain microstructures. To that aim, the manuscript focusses on a detailed and very systematic analyses of a limited number of artificial grains of different shape.

The manuscript is well and thoroughly written, easy to read and unterstand, and it makes references to most recent and also most appropriate works. The simulation/analysis is well performed and described in detail.
The authors even compare and discuss their theoretical results with optical micrographs of steel that has been published recently.

Even though this work focusses on two-dimensional investigations, it presents an important step to a better understanding of complex grain morphologies and how they form.

Author Response

We want to thank this reviewer for her/his encouraging statement and for communicating his very positive opinion towards our manuscript.

Reviewer 3 Report

Authors in the current manuscript expanded their considerations on the simulation of grain growth. This work significantly complements the previously published paper. In my opinion, a more extensive validation of theoretical results (for example, for different steel grades the exhibiting the various nature of grain growth) could be useful. spelling, grammar: generally, commas and articles are missing; Line 34. contributes to minimize - minimizing; Line 21. A strong elongation of several grains are predicted - is; Line 99. latter models - probably later; Line 143. inititial length - initial; Line 410. occuring - occurring; etc.

Author Response

We want to thank this reviewer for his work on our manuscript providing us with valuable remarks. The reviewer argues that ”a more extensive validation for the theoretical results (for example, for different steel grades the exhibiting the various nature of grain growth)  could be useful”.

We tried to state clearly that the effective triple junction and grain boundary mobilities are parameters to simulate solute drag and grain boundary pinning during the grain growth mechanism and we added the following statement: “Comparable to the results of the computer simulations, heavily distorted grains, straight grain boundaries close to strongly curved grain boundaries and strongly elongated grains can be frequently found in austenite grain structures of micro-alloyed line pipe steels as well (see e.g. [50]). It is expected in these steels that the effective mobility of grain boundaries and triple junctions is small and inhomogeneously distributed due to solute drag and particle pinning.”

Thank you also for indicating some errors and typos, which have been corrected.